# Confirmation of HLA-II associations with TB susceptibility in admixed African samples

Dayna Adrienne Croock[1], Yolandi Swart[1], Haiko Schurz[1], Desiree C Petersen[1], Marlo Möller[1,2], Caitlin Uren[1,2]*

[1]DSI-NRF Centre of Excellence for Biomedical Tuberculosis Research, South African Medical Research Council Centre for Tuberculosis Research, Division of Molecular Biology and Human Genetics, Faculty of Medicine and Health Sciences, Stellenbosch University, Stellenbosch, South Africa; [2]Centre for Bioinformatics and Computational Biology, Stellenbosch University, Stellenbosch, South Africa

## eLife Assessment

This **valuable** study confirms the association between the human leukocyte antigen (HLA)-II region and tuberculosis (TB) susceptibility in genetically admixed South African populations, specifically identifying a near-genome-wide significant association in the HLA-DPB1 gene, which originates from KhoeSan ancestry. The evidence supporting the association between the HLA-II region and TB susceptibility is **solid**, and the work will be of interest to those studying the genetic basis of tuberculosis susceptibility/infection resistance.

**\*For correspondence:**
caitlinu@sun.ac.za

**Competing interest:** The authors declare that no competing interests exist.

**Abstract** Previously, the International Tuberculosis Host Genetics Consortium (ITHGC) demonstrated the power of large-scale GWAS analysis across diverse ancestries in identifying tuberculosis (TB) susceptibility loci (Schurz et al., 2024). Despite identifying a significant genetic correlate in the human leukocyte antigen (HLA)-II region, this association did not replicate in the African ancestry-specific analysis, due to small sample size and the inclusion of admixed samples. Our study aimed to build upon the findings from the ITHGC and identify TB susceptibility loci in an admixed South African cohort using the local ancestry allelic adjusted association (LAAA) model. We identified a suggestive association peak (*rs3117230*, p-value = $5.292 \times 10^{-6}$, OR = 0.437, SE = 0.182) in the *HLA-DPB1* gene originating from KhoeSan ancestry. These findings extend the work of the ITHGC, underscore the need for innovative strategies in studying complex admixed populations, and confirm the role of the HLA-II region in TB susceptibility in admixed South African samples.

## Introduction

Tuberculosis (TB) is a communicable disease caused by *Mycobacterium tuberculosis* (*M.tb*) (*World Health Organization, 2023*). *M.tb* infection has a wide range of clinical manifestations from asymptomatic, non-transmissible, or so-called 'latent', infections to active TB (*Zaidi et al., 2023*). Approximately 1/4 of the global population is infected with *M.tb*, but only 5–15% of infected individuals will develop active TB (*Menzies et al., 2021*). Several factors increase the risk of progressing to active TB, including co-infection with HIV and comorbidities, such as diabetes mellitus, asthma and other airway and lung diseases (*Glaziou et al., 2018*). Socio-economic factors including smoking, malnutrition, alcohol abuse, intravenous drug use, prolonged residence in a high burdened community, overcrowding, informal housing and poor sanitation also influence *M.tb* transmission and infection

(*Cudahy et al., 2020*; *Escombe et al., 2019*; *Laghari et al., 2019*; *Matose et al., 2019*; *Smith et al., 2023*). Additionally, individual variability in infection and disease progression has been attributed to variation in the host genome (*Schurz et al., 2024*; *Uren et al., 2020*; *Verhein et al., 2018*; *Uren et al., 2021*). Numerous genome-wide association studies (GWASs) investigating TB susceptibility have been conducted across different population groups. However, findings from these studies often do not replicate across population groups (*Möller and Kinnear, 2020*; *Möller et al., 2018*; *Uren et al., 2017*). This lack of replication could be caused by small sample sizes, variation in phenotype definitions among studies, variation in linkage disequilibrium (LD) patterns across different population groups and the presence of population-specific effects (*Möller and Kinnear, 2020*). Additionally, complex LD patterns within population groups, produced by admixture, impede the detection of statistically significant loci when using traditional GWAS methods (*Swart et al., 2020*).

The International Tuberculosis Host Genetics Consortium (ITHGC) performed a meta-analysis of TB GWAS results including 14 153 TB cases and 19 536 controls of African, Asian and European ancestries (*Schurz et al., 2024*). The multi-ancestry meta-analysis identified one genome-wide significant variant (*rs28383206*) in the human leukocyte antigen (HLA)-II region (p=5.2 x 10$^{-9}$, OR = 0.89, 95% CI=0.84–0.95). The association peak at the *HLA-II* locus encompassed several genes encoding crucial antigen presentation proteins (including *HLA-DR* and *HLA-DQ*). While ancestry-specific association analyses in the European and Asian cohorts also produced suggestive peaks in the HLA-II region, the African ancestry-specific association test did not yield any significant associations or suggestive peaks. The authors described possible reasons for the lack of associations, including the smaller sample size compared to the other ancestry-specific meta-analyses, increased genetic diversity within African individuals and population stratification produced by two admixed cohorts from the South African Coloured (SAC) population (*Schurz et al., 2024*). The SAC population (as termed in the South African census *Lehohla, 2012*) forms part of a multi-way (up to five-way) admixed population with ancestral contributions from Bantu-speaking African (~30%), KhoeSan (~30%), European (~20%), and East (~10%) and Southeast Asian (~10%) populations (*Chimusa et al., 2013*). The diverse genetic background of admixed individuals can lead to population stratification, potentially introducing confounding variables. However, the power to detect statistically significant loci in admixed populations can be improved by leveraging admixture-induced local ancestry (*Swart et al., 2021*; *Swart et al., 2022a*). Since previous computational algorithms did not include local ancestry as a covariate for GWASs, the local ancestry allelic adjusted association model (LAAA) was developed to overcome this limitation (*Duan et al., 2018*). The LAAA model identifies ancestry-specific alleles associated with the phenotype by including the minor alleles and the corresponding ancestry of the minor alleles (obtained by local ancestry inference) as covariates. The LAAA model has been successfully applied in a cohort of multi-way admixed SAC individuals to identify novel variants associated with TB susceptibility (*Swart et al., 2021*; *Swart et al., 2022b*).

Our study builds upon the findings from the ITHGC (*Schurz et al., 2024*) and aims to resolve the challenges faced in African ancestry-specific association analysis. Here, we explore host genetic correlates of TB in a complex admixed SAC population using the LAAA model.

## Results
### Global and local ancestry inference
After close inspection of global ancestry proportions generated using ADMIXTURE, the K number of contributing ancestries (the lowest k-value determined through cross-validation) was K=3 for the

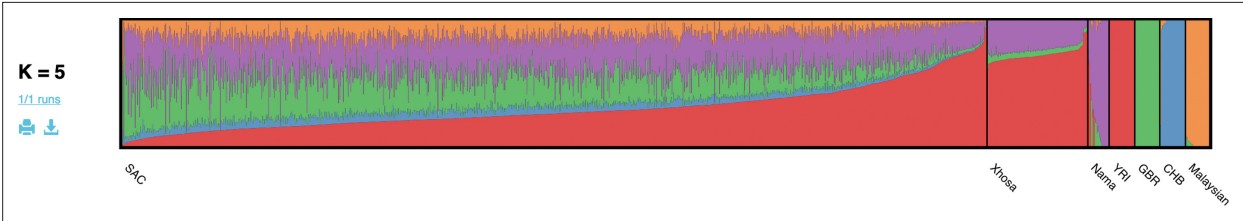

**Figure 1.** Genome-wide ancestral proportions of all individuals in the merged dataset. Ancestral proportions for each individual are plotted vertically with different colours representing different contributing ancestries.

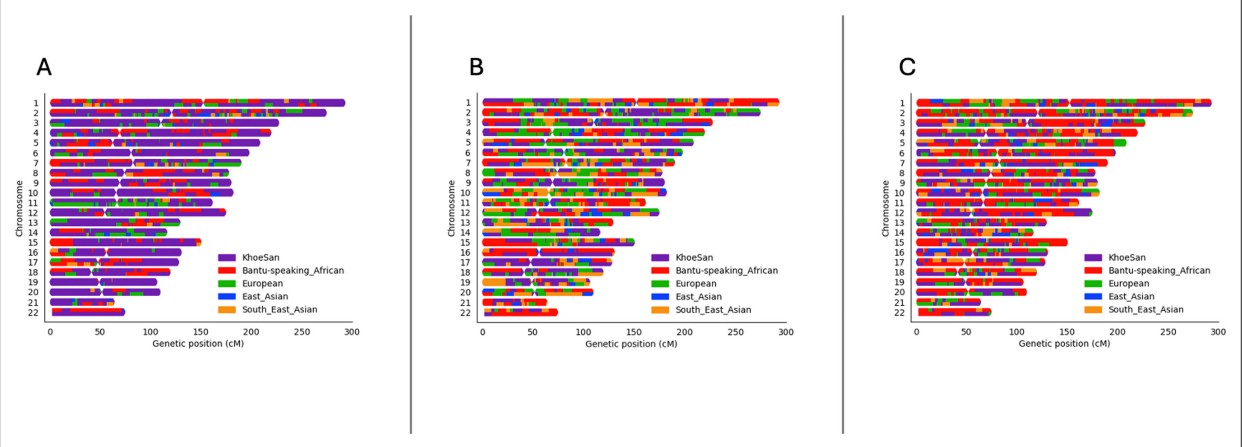

**Figure 2.** Local ancestry karyograms of three admixed individuals from the SAC population. Each admixed individual (**A, B and C**) has unique local ancestry patterns generated by admixture among geographically distinct ancestral population groups.

Xhosa individuals and K=5 for the SAC individuals (*Figure 1*). This is consistent with previous global ancestry deconvolution results (*Chimusa et al., 2014*; *Choudhury et al., 2021*). It is evident that our cohort is a complex, highly admixed group with ancestral contributions from the indigenous KhoeSan (~22–30%), Bantu-speaking African (~30–72%), European (~5–24%), Southeast Asian (~11%), and East Asian (~5%) population groups.

Local ancestry was estimated for all individuals. Admixture between geographically distinct populations creates complex ancestral and admixture-induced LD blocks, which can be visualised using local ancestry karyograms. *Figure 2* shows karyograms for three individuals from the merged dataset. It is evident that, despite individuals being from the same population group, each possesses unique patterns of local ancestry arising from differing numbers and lengths of ancestral segments.

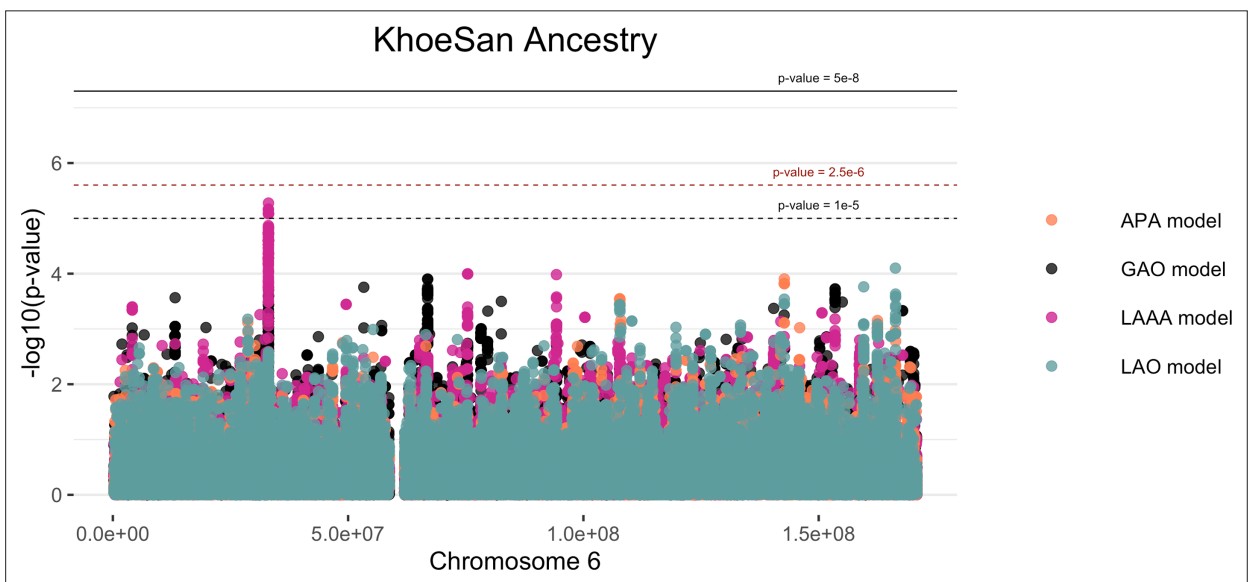

**Figure 3.** Log transformation of association signals obtained for KhoeSan ancestry whilst using the LAAA model on chromosome 6. The thresholds for genome-wide significance (p-value = 5 x 10⁻⁸) and suggestive significance (p-value = 1 x 10⁻⁵) and the significance threshold for admixture mapping (p-value = 2.5 x 10⁻⁶) are shown. The four different models are represented in black (global ancestry only - GAO), blue (local ancestry effect - LAO), orange (ancestry plus allelic effect - APA), and pink (local ancestry adjusted allelic effect - LAAA).

The online version of this article includes the following figure supplement(s) for figure 3:

**Figure supplement 1.** QQ-plot of expected p-values and observed p-values for the association signals obtained for Khoisan ancestry located on chromosome 6.

## Local ancestry-allelic adjusted analysis

LAAA models were successfully applied for all five contributing ancestries (KhoeSan, Bantu-speaking African, European, East Asian and Southeast Asian). However, no variants passed the threshold for statistical significance. Although no variants reached genome-wide significance, a suggestive peak was identified in the HLA-II region of chromosome 6 when using the LAAA model and adjusting for KhoeSan ancestry (*Figure 3*). The QQ-plot suggested minimal genomic inflation, which was verified by calculating the genomic inflation factor ($\lambda$ =1.05289; *Figure 3—figure supplement 1*). The lead variants identified using the LAAA model whilst adjusting for KhoeSan ancestry in this region on chromosome 6 are summarised in *Table 1*. The suggestive peak encompasses the *HLA-DPA1/B1* (major histocompatibility complex, class II, DP alpha 1/beta 1) genes (*Figure 4*). It is noteworthy that without the LAAA model, this suggestive peak would not have been observed for this cohort. This highlights the importance of utilising the LAAA model in future association studies when investigating disease susceptibility loci in admixed individuals, such as the SAC population.

The lead variant within this suggestive peak lies within *COL11A2P1* (collagen type X1 alpha 2 pseudogene 1). *COL11A2P1* is an unprocessed pseudogene (ENSG00000228688). Unprocessed pseudogenes are seldom transcribed and translated into functional proteins (*Witek and Mohiuddin, 2024*). *HLA-DPB1* and *HLA-DPA1* are the closest functional protein-coding genes to our lead variants. The lead variant identified in the ITHGC meta-analysis, *rs28383206*, was not present in our genotype or imputed datasets. The ITHGC imputed genotypes using the 1000 Genomes (1000 G) reference panel (*Schurz et al., 2024*). The lead variant, *rs28383206,* has an alternate allele frequency of 11.26% in the African population subgroup within the 1000 G dataset (https://www.ncbi.nlm.nih.gov/snp/rs28383206). However, *rs28383206* is absent from our in-house whole-genome sequencing (WGS) datasets, which include Bantu-speaking African and KhoeSan individuals. This absence suggests that *rs28383206* might not have been imputed in our datasets using the AGR reference panel, potentially due to its low alternate allele frequency in southern African populations. Our merged dataset contained two variants located within 800 base pairs of *rs28383206: rs482205* (6:32576009) and *rs482162* (6:32576019). However, these variants were not significantly associated with TB status in our cohort (*Supplementary file 1*).

## Discussion

The LAAA analysis of host genetic susceptibility to TB, involving 942 TB cases and 592 controls, identified one suggestive association peak adjusting for KhoeSan local ancestry. The association peak identified in this study encompasses the *HLA-DPB1* gene, a highly polymorphic locus, with over 2000 documented allelic variants (*Robinson et al., 2020*). This association is noteworthy given that *HLA-DPB1* alleles have been associated with TB resistance (*Dawkins et al., 2022*; *Ravikumar et al., 1999*; *Selvaraj et al., 2008*). The direction of effect of the lead variants in our study (*Table 1*) similarly suggests a protective effect against developing active TB. However, variants in *HLA-DPB1* were not identified in the ITHGC meta-analysis.

The ITHGC did not identify any significant associations or suggestive peaks in their African ancestry-specific analyses. Notably, the suggestive peak in the *HLA-DPB1* region was only captured in our cohort using the LAAA model whilst adjusting for KhoeSan local ancestry. This underscores the importance of incorporating global and local ancestry in association studies investigating complex multi-way admixed individuals, as the genetic heterogeneity present in admixed individuals (produced as a result of admixture-induced and ancestral LD patterns) may cause association signals to be missed when using traditional association models (*Duan et al., 2018*; *Swart et al., 2022b*).

We did not replicate the significant association signal in *HLA-DRB1* identified by the ITHGC. However, the ITHGC also did not replicate this association in their own African ancestry-specific analysis. The significant association, *rs28383206*, identified by the ITHGC meta-analysis appears to be tagging the *HLA-DQA1*02:1* allele, which is associated with TB in Icelandic and Asian populations (*Li et al., 2021*; *Sveinbjornsson et al., 2016*; *Zheng et al., 2018*). It is possible that this association signal is specific to non-African populations, but additional research is required to verify this hypothesis. Both our study and the ITHGC independently pinpointed variants associated with TB susceptibility in different genes within the HLA-II locus (*Figure 5*). The HLA-II region spans ~0.8 Mb on chromosome 6p21.32 and encompasses the *HLA-DP, -DR,* and *-DQ* alpha and beta chain genes. The

**Table 1.** Suggestive associations (p-value <1e$^{-5}$) for the LAAA analysis adjusting for KhoeSan local ancestry on chromosome 6.

| Position | Marker name | Ref | Alt | AltFreq | OR (95% CI) | SE | p-value (x10$^{-6}$) | Gene | Location | Imputed/typed | INFO score |
|---|---|---|---|---|---|---|---|---|---|---|---|
| 33075635 | rs3117230 | A | G | 0.370 | 0.437 (0.306; 0.624) | 0.182 | 5.292 | HLA-DPB1 | Intergenic | Genotyped | NA |
| 33048661 | rs1042151 | A | G | 0.325 | 0.437 (0.305; 0.627) | 0.184 | 6.806 | HLA-DPB1 | Exonic | Imputed | 0.992 |
| 33058874 | rs2179920 | C | T | 0.369 | 0.445 (0.313; 0.633) | 0.180 | 6.960 | HLA-DPB1 | Intergenic | Genotyped | NA |
| 33072266 | rs2064478 | C | T | 0.371 | 0.447 (0.313; 0.637) | 0.181 | 8.222 | HLA-DPB1 | Intergenic | Imputed | 1 |
| 33072729 | rs3130210 | G | T | 0.371 | 0.447 (0.313; 0.637) | 0.181 | 8.222 | HLA-DPB1 | Intergenic | Imputed | 0.999 |
| 33073440 | rs2064475 | G | A | 0.371 | 0.447 (0.313; 0.637) | 0.181 | 8.222 | HLA-DPB1 | Intergenic | Imputed | 1 |
| 33074348 | rs3117233 | T | C | 0.371 | 0.447 (0.313; 0.637) | 0.181 | 8.222 | HLA-DPB1 | Intergenic | Imputed | 1 |
| 33074707 | rs3130213 | G | A | 0.371 | 0.447 (0.313; 0.637) | 0.181 | 8.222 | HLA-DPB1 | Intergenic | Imputed | 0.970 |

Ref, reference allele; Alt, alternate allele; AltFreq, alternate allele frequency; OR, odds ratio; SE, standard error.

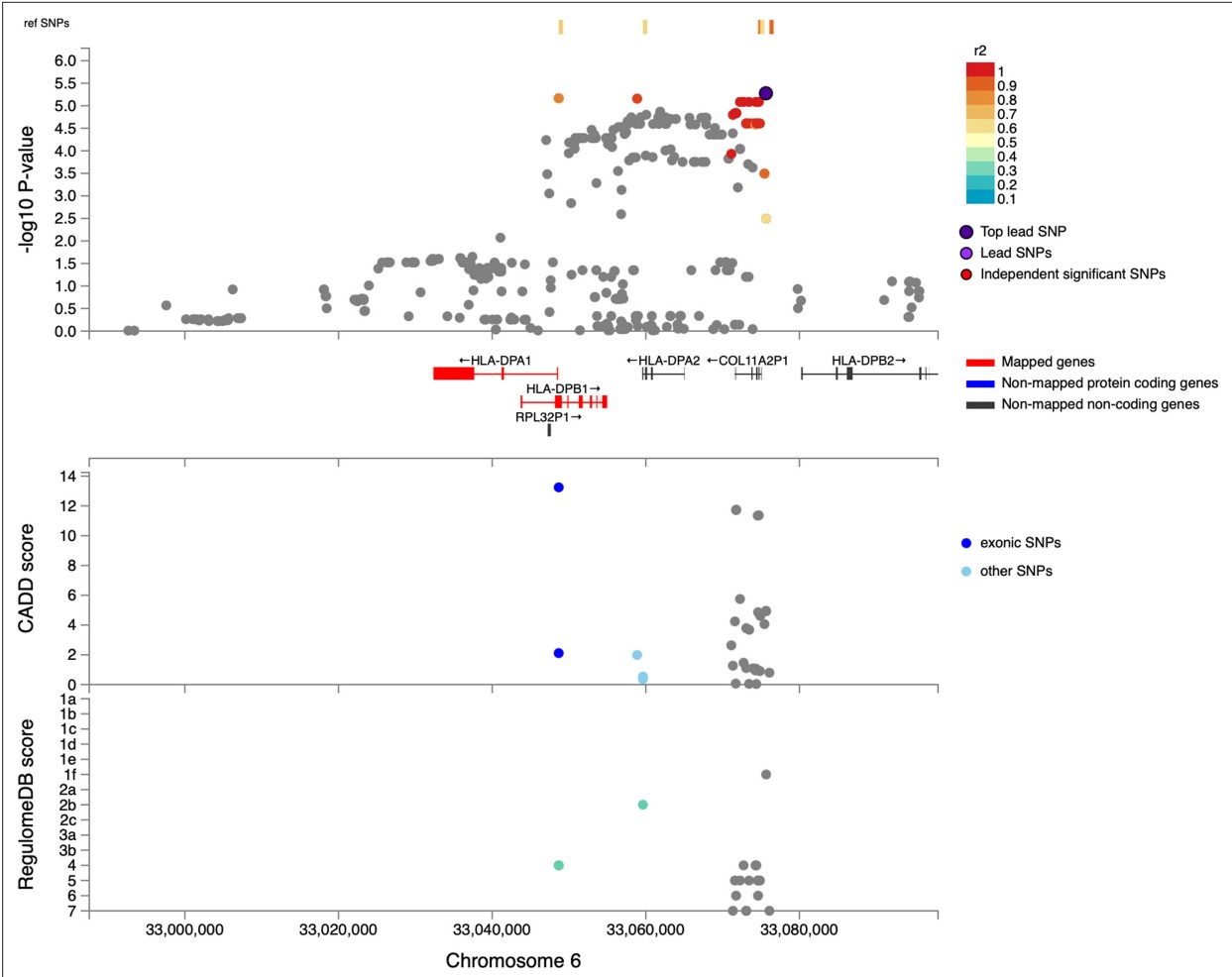

**Figure 4.** Regional plot indicating the nearest genes in the region of the lead variant (*rs3117230*) observed on chromosome 6. SNPs in linkage disequilibrium (LD) with the lead variant are coloured red/orange. The lead variant is indicated in purple. Functional protein-coding genes are coded in red and non-functional (pseudo-genes) are indicated in black.

HLA-II complex is the human form of the major histocompatibility complex class II (MHC-II) proteins on the surface of antigen presenting cells, such as monocytes, dendritic cells and macrophages. The innate immune response against *M.tb* involves phagocytosis by alveolar macrophages. In the phagosome, mycobacterial antigens are processed for presentation on MHC-II on the surface of the antigen presenting cell. Previous studies have suggested that *M.tb* interferes with the MHC-II pathway to enhance intracellular persistence and delay activation of the adaptive immune response (***Oliveira-Cortez et al., 2016***). For example, *M.tb* can inhibit phagosome maturation and acidification, thereby limiting antigen processing and presentation on MHC-II molecules (***Chang et al., 2005***). Given that MHC-II plays an essential role in the adaptive immune response to TB and numerous studies have identified HLA-II variants associated with TB (***Cai et al., 2019***; ***Chihab et al., 2023***; de ***de Sá et al., 2020***; ***Harishankar et al., 2018***; ***Schurz et al., 2024***; ***Selvaraj et al., 2008***), additional research is required to elucidate the effects of HLA-II variation on TB risk status.

This analysis has a few limitations. First, unlike the ITHGC manuscript, we did not validate our SNP peak in the HLA-II region through fine mapping. Although we initially considered performing HLA imputation and fine-mapping using the HIBAG R package, as described in the ITHGC article (https://hibag.s3.amazonaws.com/hlares_index.html#estimates), the African HIBAG model was trained on genotype data from African American and HapMap YRI populations, which have minimal to no KhoeSan ancestry. Since our association peak likely originates from KhoeSan ancestral haplotype blocks, using an imputation reference panel that includes individuals with KhoeSan ancestry is essential to this analysis. We acknowledge that HLA typing could validate the importance of our lead SNPs

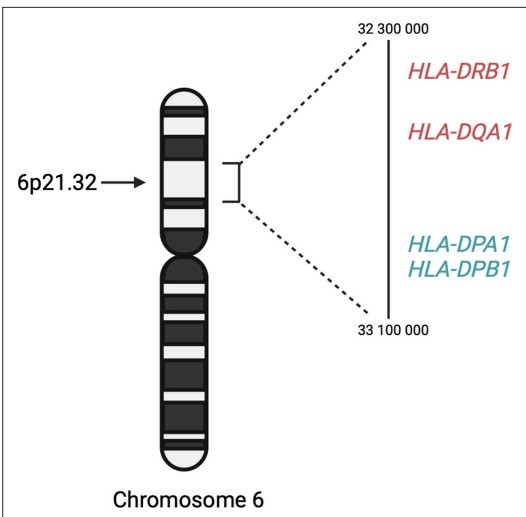

**Figure 5.** A schematic diagram of the location of HLA-II genes associated with TB susceptibility. Genes in red were identified by the ITHGC. Genes in blue were identified by this study.

in the HLA-II region and support the LAAA model, but this was not feasible due to the absence of a suitable reference panel that includes KhoeSan ancestry. Second, our analysis has a notable case-control imbalance (cases/controls = 1.610). While many studies discuss methods for addressing case-control imbalances with more controls than cases which can inflate type 1 error rates (*Dai et al., 2021*; *Öztornaci et al., 2023*; *Zhou et al., 2018*), few address the implications of a large case-to-control ratio like ours (952 cases to 592 controls). To assess the impact of this imbalance, we used the Michigan genetic association study (GAS) power calculator (*Skol et al., 2006*). Under an additive disease model with an estimated prevalence of 0.15, a disease allele frequency of 0.3, a genotype relative risk of 1.5, and a default significance level of $7 \times 10^{-6}$, we achieved an expected power of approximately 75%. With a balanced sample size of 950 cases and 950 controls, power would exceed 90%, but it would drop significantly with a smaller balanced cohort of 590 cases and 590 controls. Given these results, we proceeded with our analysis to maximise statistical power despite the case-control imbalance.

In conclusion, the application of the LAAA to a highly admixed SAC cohort revealed a suggestive association signal in the HLA-II region associated with protection against TB that was not identified by the African-ancestry specific analysis performed by the ITHGC. Our study builds on the results of the ITHGC by demonstrating an alternative method to identify association signals in cohorts with complex genetic ancestry. This analysis shows the value of including individual global and local ancestry in genetic association analyses. Furthermore, we confirm HLA-II loci associations with TB susceptibility in an admixed South African population, highlighting the role of the adaptive immune system in TB susceptibility and resistance.

## Materials and methods
### Data
This study included the two SAC admixed datasets from the ITHGC analysis [RSA(A) and RSA(M)] as well as four additional TB case-control datasets obtained from admixed South African population groups (*Table 2*). Like the SAC population, the Xhosa population is admixed with Bantu-speaking African and KhoeSan ancestral contributions (*Choudhury et al., 2021*). All datasets were collected

**Table 2.** Summary of the datasets included in analysis.

| Dataset | Genotyping platform | Self-reported ethnicity | Cases/controls | Reference |
|---|---|---|---|---|
| **RSA(A)** | Affymetrix 500 k | SAC | 642/91 | *Daya et al., 2013* |
| **RSA(M)** | MEGA array 1.1 M | SAC | 555/440 | *Schurz et al., 2018*; *Swart et al., 2021* |
| **RSA(TANDEM)** | H3Africa array | SAC and Bantu-speaking African | 161/133 | *Swart et al., 2022b* |
| **RSA(NCTB)** | H3Africa array | SAC | 49/111 | *Oyageshio et al., 2023* |
| **RSA(Worcester)** | H3Africa array | SAC | 61 cases | Unpublished |
| **RSA(Xhosa)** | Whole genome sequencing | IsiXhosa | 44/120 | Unpublished |

over the past 30 years under different research projects (*Daya et al., 2013*; *Kroon et al., 2020*; *Schurz et al., 2018*; *Smith et al., 2023*; *Ugarte-Gil et al., 2020*) and individuals that were included in the analyses consented to the use of their data in future research regarding TB host genetics. Across all datasets, TB cases were bacteriologically confirmed (culture positive) or diagnosed by GeneXpert. Controls were healthy individuals with no history of TB disease or treatment. However, given the high prevalence of TB in South Africa 852 cases (95% CI 679–1026) per 10,000 individuals 15 years and older (*Cudahy et al., 2020*), most controls have likely been exposed to *M.tb* at some point (*Gallant et al., 2010*). For all datasets, cases and controls were obtained from the same community and thus share similar socio-economic status and health care access.

A list of sites genotyped on the Infinium H3Africa array (https://chipinfo.h3abionet.org/browse) was extracted from the whole-genome sequenced [RSA(Xhosa)] dataset and treated as genotype data in subsequent analyses. Quality control (QC) of raw genotype data was performed using PLINK v1.9 (*Purcell et al., 2007*). In all datasets, individuals were screened for sex concordance and discordant sex information was corrected based on X chromosome homozygosity estimates ($F_{estimate}$ <0.2 for females and $F_{estimate}$ >0.8 for males). In the event that sex information could not be corrected based on homozygosity estimates, individuals with missing or discordant sex information were removed. Individuals with genotype call rates less than 90% and SNPs with more than 5% missingness were removed as described previously (*Swart et al., 2021*). Monomorphic sites were removed. Individuals were screened for deviations in Hardy-Weinberg Equilibrium (HWE) for each SNP, and sites deviating from the HWE threshold of $10^{-5}$ were removed. Sex chromosomes were excluded from the analysis. The genome coordinates across all datasets were checked for consistency and, if necessary, converted to GRCh37 using the UCSC liftOver tool (*Kuhn et al., 2013*). The number of individuals and variants remaining after genotype QC is shown in *Supplementary file 2*.

Genotype datasets were pre-phased using SHAPEIT v2 (*Delaneau et al., 2013*) and imputed using the Positional Burrows-Wheeler Transformation (PBWT) algorithm through the Sanger Imputation Server (SIS; *Durbin, 2014*). The African Genome Resource (AGR) panel (n=4956), accessed via the SIS, was used as the reference panel for imputation (*Gurdasani et al., 2015*) since it has been shown that the AGR is the best reference panel for imputation of missing genotypes for samples from the SAC population (*Schurz et al., 2019*). Imputed data were filtered to remove sites with imputation quality INFO scores less than 0.95. Individual datasets were screened for relatedness using KING software (*Manichaikul et al., 2010*) and individuals up to second degree relatedness were removed. A total of 7,544,769 markers overlapped across all six datasets. This list of intersecting markers was extracted from each dataset using the PLINK `--extract` flag. The datasets were then merged using the PLINK v1.9. After merging, all individuals missing more than 10% genotypes were removed, markers with more than 5% missing data were excluded and a HWE filter was applied to controls (threshold $<10^{-5}$). The merged dataset was screened for relatedness using KING, and individuals up to second degree relatedness were subsequently removed. The final merged dataset after QC and data filtering (including the removal of related individuals) consisted of 1 544 individuals (952 TB cases and 592 healthy controls). A total of 7,510,057 variants passed QC and filtering parameters.

## Global ancestry inference

ADMIXTURE was used to determine the correct number of contributing ancestral proportions in our multi-way admixed population cohort (*Alexander and Lange, 2011*). ADMIXTURE estimates the number of contributing ancestral populations (denoted by K) and population allele frequencies through cross-validation (CV). All 1544 individuals were grouped into running groups of equal size

**Table 3.** Ancestral populations included for global ancestry deconvolution.

| Population | n | Source |
|---|---|---|
| European (British – GBR) | 40 | 1000 Genomes (1000 G) phase 3 (*Auton et al., 2015*) |
| East Asian (Chinese – CHB) | 40 | 1000 G phase 3 |
| Bantu-speaking African (Yoruba – YRI) | 40 | 1000 G phase 3 |
| Southeast Asian (Malaysian) | 38 | Singapore Sequencing Malay Project (SSMP) (*Wong et al., 2013*) |
| KhoeSan (Nama) | 33 | African Genome Variation Project (AGVP/ADRP) (*Gurdasani et al., 2015*) |

together with 191 reference populations (*Table 3*). Running groups were created to ensure approximately equal numbers of reference populations and admixed populations. Xhosa and SAC samples were divided into separate running groups.

Redundant SNPs were removed by PLINK through LD pruning by removing each SNP with LD $r^2$ >0.1 within a 50-SNP sliding window (advanced by 10 SNPs at a time). Ancestral proportions were inferred in an unsupervised manner for K=3–6 (1 iteration). The best value of K for the data was selected by choosing the K value with the lowest CV error across all running groups. Ten iterations of K=3 and K=5 were run for the Xhosa and SAC individuals respectively. Since it has been shown that RFMix (*Maples et al., 2013*) outperforms ADMIXTURE in determining global ancestry proportions (*Uren et al., 2020*), RFMix was also used to refine inferred global ancestry proportions. Global ancestral proportions were visualised using PONG (*Behr et al., 2016*).

### Local ancestry inference
The merged dataset and the reference file (containing reference populations from *Table 3*) were phased separately using SHAPEIT2. The local ancestry for each position in the genome was inferred using RFMix (*Maples et al., 2013*). Default parameters were used, but the number of generations since admixture was set to 15 for the SAC individuals and 20 for the Xhosa individuals (as determined by previous studies) (*Uren et al., 2016*). RFMix was run with three expectation maximisation iterations and the `--reanalyse-reference` flag.

### Batch effect screening and correction
Merging separate datasets generated at different timepoints and/or facilities, as we have done here, will undoubtedly introduce batch effects. Principal component analysis (PCA) is a common method used to visualise batch effects, where the first two principal components (PCs) are plotted with each sample coloured by batch, and a separation of colours is indicative of a batch effect (*Nyamundanda et al., 2017*). However, it is difficult to differentiate between separation caused by population structure and separation caused by batch effect using PCA alone. An alternative method to detect batch effects (*Chen et al., 2022*) involves coding case/control status by batch, followed by running an association analysis testing each batch against all other batches. If any single dataset has more positive signals compared to the other datasets, then batch effects may be responsible for producing spurious results. Batch effects can be resolved by removing those SNPs which pass the genome-wide significance threshold from the merged dataset. We have adapted this batch effect correction method for application in a highly admixed cohort with complex population structure (*Croock et al., 2024*). Code required to execute batch effect correction procedures is publicly available (https://github.com/TBHostGenetics/data_harmonisation, copy archived at *Croock, 2025*). Our modified method was used to remove 36 627 SNPs affected by batch effects from our merged dataset.

### Local ancestry allelic adjusted association analysis
The LAAA association model was used to investigate if there are allelic, ancestry-specific or ancestry-specific allelic associations with TB susceptibility in our merged dataset. Global ancestral components inferred by RFMix, age and sex were included as covariates in the association tests (*Supplementary file 3*). Variants with minor allele frequency (MAF) <1% were removed to improve the stability of the association tests. A total of 784,557 autosomal markers (with MAF >1%) and 1544 unrelated individuals (952 TB cases and 592 healthy controls) were available for further analyses. Of the markers included in the final dataset, 535,193 sites were imputed. Dosage files, which code the number of alleles of a specific ancestry at each locus across the genome, were compiled. Separate regression models for each ancestral contribution were fitted to investigate which ancestral contribution is associated with TB susceptibility. Code required to execute the LAAA model is publicly available (https://github.com/TBHostGenetics/LAAA-model, copy archived at *Swart, 2025*). Details regarding the models have been described elsewhere (*Swart et al., 2022b*); but in summary, four regression models were tested to detect the source of the association signals observed:

### (1) Null model or global ancestry (GA) model
The null model only includes global ancestry, sex and age covariates. This test investigates whether an additive allelic dose exerts an effect on the phenotype (without including local ancestry of the allele).

### (2) Local ancestry (LA) model

This model is used in admixture mapping to identify ancestry-specific variants associated with a specific phenotype. The LA model evaluates the number of alleles of a specific ancestry at a locus and includes the corresponding marginal effect as a covariate in association analyses.

### (3) Ancestry plus allelic (APA) model

The APA model simultaneously performs model (1) and (2). This model tests whether an additive allelic dose exerts an effect on the phenotype whilst adjusting for local ancestry.

### (4) Local ancestry adjusted allelic (LAAA) model

The LAAA model is an extension of the APA model, which models the combination of the minor allele and ancestry of the minor allele at a specific locus and the effect this interaction has on the phenotype.

The R package *STEAM* (Significance Threshold Estimation for Admixture Mapping; *Grinde et al., 2019*) was used to determine the admixture mapping significance threshold given the global ancestral proportions of each individual and the number of generations since admixture (g=15). For the LA model, a genome-wide significance threshold of p-value $<2.5 \times 10^{-6}$ was deemed significant by *STEAM*. The traditional genome-wide significance threshold of $5 \times 10^{-8}$ was used for the GA, APA and LAAA models, as recommended by the authors of the LAAA model (*Duan et al., 2018*). Results from the analysis performed on chromosome 6 whilst adjusting for KhoeSan ancestry are documented in *Supplementary file 4*.

## Acknowledgements

We acknowledge the support of the DSI-NRF Centre of Excellence for Biomedical Tuberculosis Research, South African Medical Research Council Centre for Tuberculosis Research (SAMRC CTR), Division of Molecular Biology and Human Genetics, Faculty of Medicine and Health Sciences, Stellenbosch University, Cape Town, South Africa. We also acknowledge the Centre for High Performance Computing (CHPC), South Africa, for providing computational resources. This research was partially funded by the South African government through the SAMRC and the Harry Crossley Research Foundation.

## Additional information

### Funding

| Funder | Grant reference number | Author |
| --- | --- | --- |
| South African Medical Research Council | | Dayna Adrienne Croock |
| Harry Crossley Foundation | | Dayna Adrienne Croock |

The funders had no role in study design, data collection and interpretation, or the decision to submit the work for publication.

### Author contributions

Dayna Adrienne Croock, Formal analysis, Investigation, Visualization, Methodology, Writing – original draft, Writing – review and editing; Yolandi Swart, Resources, Supervision, Methodology, Writing – review and editing; Haiko Schurz, Conceptualization, Supervision, Methodology, Writing – review and editing; Desiree C Petersen, Conceptualization, Supervision, Writing – review and editing; Marlo Möller, Conceptualization, Data curation, Supervision, Writing – review and editing; Caitlin Uren, Conceptualization, Resources, Data curation, Supervision, Project administration, Writing – review and editing

### Author ORCIDs

Dayna Adrienne Croock ![ORCID] https://orcid.org/0000-0002-5107-8006
Haiko Schurz ![ORCID] https://orcid.org/0000-0002-0009-3409
Desiree C Petersen ![ORCID] https://orcid.org/0000-0002-0817-2574

Marlo Möller ⓘ https://orcid.org/0000-0002-0805-6741
Caitlin Uren ⓘ http://orcid.org/0000-0003-2358-0135

### Ethics

Ethics approval was granted by the Health Research Ethics Committee (HREC) of Stellenbosch University, South Africa (project number S22/02/031). Individuals that were included in the analyses consented to the use of their data in future research regarding TB host genetics.

Reviewer #2 (Public review): https://doi.org/10.7554/eLife.99200.4.sa1
Author response https://doi.org/10.7554/eLife.99200.4.sa2

---

## Additional files

### Supplementary files

MDAR checklist

Supplementary file 1. Summary statistics for two variants within 800 base pairs of the ITHGC lead SNP 167 (rs28383206) on chromosome 6 for the LAAA analysis adjusting for KhoeSan and Bantu-speaking African local 168 ancestry.

Supplementary file 2. The number of individuals and variants across all array datasets following genotype QC.

Supplementary file 3. Summary of the age, sex and ancestral proportions for individuals in the merged cohort.

Supplementary file 4. Summary statistics of the results for chromosome 6 whilst using the local ancestry adjusted allelic (LAAA) model whilst adjusting for KhoeSan ancestry.

### Data availability

The current manuscript is a computational study, so no new genetic data was generated for this manuscript. Access to retrospective genetic datasets analysed can be requested through the original studies data access process. Where the dataset is yet to be published, access to these datasets will be considered upon reasonable request in line with the initial participant consent - please email caitlinu@sun.ac.za. Summary statistics for the covariate data for individuals in the cohort are available in Supplementary File 3, and LAAA model results for chromosome 6 (adjusted for KhoeSan ancestry) are available in Supplementary File 4. Code required to perform genotype QC, imputation, ancestry inference and batch effect procedures is publicly available (https://github.com/TBHostGenetics/data_harmonisation copy archived at *Croock, 2025*). Code required to execute the LAAA model is publicly available (https://github.com/TBHostGenetics/LAAA-model copy archived at *Swart, 2025*).

The following previously published dataset was used:

| Author(s) | Year | Dataset title | Dataset URL | Database and Identifier |
|---|---|---|---|---|
| Oyageshio OP, Myrick JW, Saayman J, van der Westhuizen L, Al-Hindi D, Reynolds AW, Zaitlen N, Uren C, Möller M, Henn BM | 2023 | Investigating Host Genetic Risk Factors for Tuberculosis in Highly Endemic South African Populations | https://ega-archive.org/studies/EGAS00001007850 | European Genome-Phenome Archive, EGAS00001007850 |

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
