## [Editor Report · eLife Assessment]

This **valuable** study confirms the association between the human leukocyte antigen (HLA)-II region and tuberculosis (TB) susceptibility in genetically admixed South African populations, specifically identifying a near-genome-wide significant association in the HLA-DPB1 gene, which originates from KhoeSan ancestry. The evidence supporting the association between the HLA-II region and TB susceptibility is **solid**, and the work will be of interest to those studying the genetic basis of tuberculosis susceptibility/infection resistance.

---

## [Referee Report · Reviewer #2 (Public review)]

Summary:

This manuscript is about using different analytical approaches to allow ancestry adjustments to GWAS analyses amongst admixed populations. This work is a follow-on from the recently published ITHGC multi-population GWAS (https://doi.org/10.7554/eLife.84394), with the focus on the admixed South African populations. Ancestry adjustment models detected a peak of SNPs in the class II HLA DPB1, distinct from the class II HLA DQA1 loci signficant in the ITHGC analysis.

Strengths:

Excellent demonstration of GWAS analytical pipelines in highly admixed populations. Particularly the utility of ancestry adjustment to improve study power to detect novel associations. Further confirmation of the importance of the HLA class II locus in genetic susceptibility to TB.

Weaknesses:

Limited novelty compared to the group's previous existing publications and the body of work linking HLA class II alleles with TB susceptibility in South Africa or other African populations. This work includes only ~100 new cases and controls from what has already been published. High resolution HLA typing has detected significant signals in both the DQA1 and DPB1 regions identified by the larger ITHGC and in this GWAS analysis respectively (Chihab L et al. HLA. 2023 Feb; 101(2): 124-137).

Despite the availability of strong methods for imputing HLA from GWAS data (Karnes J et Plos One 2017), the authors did not confirm with HLA typing the importance of their SNP peak in the class II region. This would have supported the importance of this ancestry adjustment versus prior ITHGC analysis.

The populations consider active TB and healthy controls (from high-burden presumed exposed communities) and do not provide QFT or other data to identify latent TB infection.

Important methodological points for clarification and for readers to be aware of when reading this paper:

(1) One of the reasons cited for the lack of African ancestry-specific associations or suggestive peaks in the ITHGC study was the small African sample size. The current association test includes a larger African cohort and yields a near-genome-wide significant threshold in the HLA-DPB1 gene originating from the KhoeSan ancestry. Investigation is needed as to whether the increase in power is due to increased African samples and not necessarily the use of the LAAA model as stated on lines 295 and 296?

Authors response - The Manhattan plot in Figure 3 includes the results for all four models: the traditional GWAS model (GAO), the admixture mapping model (LAO), the ancestry plus allelic (APA) model and the LAAA model. In this figure, it is evident that only the LAAA model identified the association peak on chromosome 6, which lends support the argument that the increase in power is due to the use of the LAAA model and not solely due to the increase in sample size.

Reviewer comment - This data supports the authors conclusions that increase power is related to the LAAA model application rather than simply increase sample size.

(2) In line 256, the number of SNPs included in the LAAA analysis was 784,557 autosomal markers; the number of SNPs after quality control of the imputed dataset was 7,510,051 SNPs (line 142). It is not clear how or why ~90% of the SNPs were removed. This needs clarification.

Authors response:

In our manuscript (line 194), we mention that "...variants with minor allele frequency (MAF) < 1% were removed to improve the stability of the association tests." A large proportion of imputed variants fell below this MAF threshold and were subsequently excluded from this analysis.

Reviewers additional comment: The authors should specify the number of SNPs in the dataset before imputation and indicate what proportion of the 784,557 remaining SNPs were imputed. Providing this information might help the reader better understand the rationale behind the imputation process.

(3) The authors have used the significance threshold estimated by the STEAM p-value < 2.5x10-6 in the LAAA analysis. Grinde et al. 2019 implemented their significance threshold estimation approach tailored to admixture mapping (local ancestry (LA) model), where there is a reduction in testing burden. The authors should justify why this threshold would apply to the LAAA model (a joint genotype and ancestry approach).

Authors response: We describe in the methods (line 189 onwards) that the LAAA model is an extension of the APA model. Since the APA model itself simultaneously performs the null global ancestry only model and the local ancestry model (utilised in admixture mapping), we thus considered the use of a threshold tailored to admixture mapping appropriate for the LAAA model.

Reviewers additional comment: While the LAAA model is an extension of the APA model, the authors describe the LAAA test as 'models the combination of the minor allele and the ancestry of the minor allele at a specific locus, along with the effect of this interaction,' thus a joint allele and ancestry effects model. Grinde et al. (2019) proposed the significance threshold estimation approach, STEAM, specifically for the LA approach, which tests for ancestry effects alone and benefits from the reduced testing burden. However, it remains unclear why the authors found it appropriate to apply STEAM to the LAAA model, a joint test for both allele and ancestry effects, which does not benefit from the same reduction in testing burden.

(4) Batch effect screening and correction (line 174) is a quality control check. This section is discussed after global and local ancestry inferences in the methods. Was this QC step conducted after the inferencing? If so, the authors should justify how the removed SNPs due to the batch effect did not affect the global and local ancestry inferences or should order the methods section correctly to avoid confusion.

Authors response: The batch effect correction method utilised a pseudo-case-control comparison which included global ancestry proportions. Thus, batch effect correction was conducted after ancestry inference. We excluded 36 627 SNPs that were believed to have been affected by the batch effect. We have amended line 186 to include the exact number of SNPs excluded due to batch effect.

The ancestry inference by RFMix utilised the entire merged dataset of 7 510 051 SNPs. Thus, the SNPs removed due to the batch effect make up a very small proportion of the SNPs used to conduct global and local ancestry inferences (less than 0.5%). As a result, we do not believe that the removed SNPs would have significantly affected the global and local ancestry inferences. However, we did conduct global ancestry inference with RFMix on each separate dataset as a sanity check. In the tables below, we show the average global ancestry proportions inferred for each separate dataset, the average global ancestry proportions across all datasets and the average global ancestry proportions inferred using the merged dataset. The SAC and Xhosa cohorts are shown in two separate tables due to the different number of contributing ancestral populations to each cohort. The differences between the combined average global ancestry proportions across the separate cohorts does not differ significantly to the global ancestry proportions inferred using the merged dataset.

This is an excellent response and should remain accessible to readers for clarifying this issue.

Comments on revisions:

Thank you for addressing my other recommendations to authors. These have all been satisfactorily addressed.

---

## [Author Response]

The following is the authors’ response to the previous reviews

**Recommendations for the authors:**

**Reviewer #1:**
First, I thank the authors for clarifying some of the confusion I had in the previous comment and I appreciate the efforts the authors put into improving the quality of the manuscript. However, my concerns about the lack of novelty of the key findings are not perfectly addressed and there is no additional analysis done in this revision. Currently in this version of the manuscript, asserting that a p-value of 10-6 is close to genome-wide significance may be considered an overstatement. Further analysis focusing on finding novel and additional discovery is very necessary.

We thank the reviewer for their comments. Reviewer #2 also made a comment regarding the genomewide threshold, “However, it remains unclear why the authors found it appropriate to apply STEAM to the LAAA model, a joint test for both allele and ancestry effects, which does not benefit from the same reduction in testing burden.” The reviewers’ have correctly identified our oversight - we have amended the manuscript as follows:

(1) The abstract, “We identified a suggestive association peak (rs3117230, p-value = 5.292 x10-6, OR = 0.437, SE = 0.182) in the HLA-DPB1 gene originating from KhoeSan ancestry.”

(2) From line 233 to 239: “The R package STEAM (Significance Threshold Estimation for Admixture Mapping) (Grinde et al., 2019) was used to determine the admixture mapping significance threshold given the global ancestral proportions of each individual and the number of generations since admixture (g = 15). For the LA model, a genome-wide significance threshold of pvalue < 2.5 x 10-6 was deemed significant by STEAM. The traditional genome-wide significance threshold of 5 x 10-8 was used for the GA, APA and LAAA models, as recommended by the authors of the LAAA model (Duan et al., 2018).”

(3) We excluded the results for the signal on chromosome 20, since this also did not reach the LAAA model genome-wide significance threshold.

(4) From line 296 to 308: “LAAA models were successfully applied for all five contributing ancestries (KhoeSan, Bantu-speaking African, European, East Asian and Southeast Asian). However, no variants passed the threshold for statistical significance. Although no variants reached genome-wide significance, a suggestive peak was identified in the HLA-II region of chromosome 6 when using the LAAA model and adjusting for KhoeSan ancestry (Figure 3). The QQ-plot suggested minimal genomic inflation, which was verified by calculating the genomic inflation factor (= 1.05289) (Supplementary Figure 1). The lead variants identified using the LAAA model whilst adjusting for KhoeSan ancestry in this region on chromosome 6 are summarised in Table 3. The suggestive peak encompasses the HLA-DPA1/B1 (major histocompatibility complex, class II, DP alpha 1/beta 1) genes (Figure 4). It is noteworthy that without the LAAA model, this suggestive peak would not have been observed for this cohort. This highlights the importance of utilising the LAAA model in future association studies when investigating disease susceptibility loci in admixed individuals, such as the SAC population.”

We acknowledge that our results are not statistically significant. However, our study advances this area of research by identifying suggestive African-specific ancestry associations with TB in the HLA-II region. These findings build upon the work of the ITHGC, which did not identify any significant associations or suggestive peaks in their African-specific analyses. We have included this argument in our manuscript (from lines 425 to 432):

“The ITHGC did not identify any significant associations or suggestive peaks in their African ancestryspecific analyses. Notably, the suggestive peak in the HLA-DPB1 region was only captured in our cohort using the LAAA model whilst adjusting for KhoeSan local ancestry. This underscores the importance of incorporating global and local ancestry in association studies investigating complex multi-way admixed individuals, as the genetic heterogeneity present in admixed individuals (produced as a result of admixtureinduced and ancestral LD patterns) may cause association signals to be missed when using traditional association models (Duan et al., 2018; Swart, van Eeden, et al., 2022).”

We appreciate the comment regarding additional analyses. We acknowledge that we did not validate our SNP peak in the HLA-II region through fine-mapping due to the lack of a suitable reference panel (see lines 490 to 500). Our long-term goal is to develop a HLA-imputation reference panel incorporating KhoeSan ancestry; however, this is beyond the scope and funding allowances of this study.

**Reviewer #2 (Recommendations for the authors):**
The authors we think have done an excellent job with their responses and the manuscript has been substantially improved.

Thank you for taking the time to help us improve our manuscript.